# Progress toward Better Treatment of Therapy-Related AML

**DOI:** 10.3390/cancers15061658

**Published:** 2023-03-08

**Authors:** Angeliki Kotsiafti, Konstantinos Giannakas, Panagiotis Christoforou, Konstantinos Liapis

**Affiliations:** 1Department of Hematology, Alexandra Hospital, 115 28 Athens, Greece; 2Department of Hematology, Metaxa Oncology Hospital, 185 37 Piraeus, Greece; 3Pathophysiology Department, National and Kapodistrian University of Athens, 157 72 Athens, Greece; 4Dragana Campus, Democritus University of Thrace Medical School, 681 00 Alexandroupolis, Greece

**Keywords:** therapy-related, acute myeloid leukemia, AML, risk stratification, individualized therapy, personalized therapy, chemotherapy, radiotherapy, alkylating agents, topoisomerase II inhibitors, venetoclax (BCL-2 inhibitor), CPX-351, *TP53* mutations, *NPM1* mutations

## Abstract

**Simple Summary:**

Therapy-related acute myeloid leukemia (t-AML) is one of the most serious long-term complications of cancer chemotherapy. Various cytotoxic agents and exposure to ionizing radiation can lead to the development of t-AML, which is usually associated with adverse genetic changes and a poor prognosis. Over the past decade, insights into leukemogenesis have generated significant advances in the risk stratification of t-AML, and have offered us the opportunity to develop individualized options for treatment that target disease biology. In this article, we review current knowledge on the biology of t-AML, putting emphasis on its molecular origin; we also discuss recent advances in its treatment including CPX-351, the use of less intensive regimens (e.g., venetoclax combined with a hypomethylating agent), and novel, molecularly targeted and antibody-based therapies that promise to increase the cure rate.

**Abstract:**

Therapy-related acute myeloid leukemia (t-AML) comprises 10–20% of all newly diagnosed cases of AML and is related to previous use of chemotherapy or ionizing radiotherapy for an unrelated malignant non-myeloid disorder or autoimmune disease. Classic examples include alkylating agents and topoisomerase II inhibitors, whereas newer targeted therapies such as poly (adenosine diphosphate–ribose) polymerase (PARP) inhibitors have emerged as causative agents. Typically, t-AML is characterized by adverse karyotypic abnormalities and molecular lesions that confer a poor prognosis. Nevertheless, there are also cases of t-AML without poor-risk features. The management of these patients remains controversial. We describe the causes and pathophysiology of t-AML, putting emphasis on its mutational heterogeneity, and present recent advances in its treatment including CPX-351, hypomethylating agent plus venetoclax combination, and novel, molecularly targeted agents that promise to improve the cure rates. Evidence supporting personalized medicine for patients with t-AML is presented, as well as the authors’ clinical recommendations.

## 1. Introduction

According to the 2017 World Health Organization (WHO) classification system for tumors of hematopoietic and lymphoid tissues, therapy-related myeloid neoplasms include cases of acute myeloid leukemia (t-AML), myelodysplastic syndromes (t-MDS), and myelodysplastic/myeloproliferative neoplasms (t-MDS/MPN), which arise as a complication of cytotoxic chemotherapy and/or radiation therapy administered for a prior neoplastic or non-neoplastic disorder [1]. Epidemiologic evidence shows that the incidence of t-AML has greatly increased during the past three decades as a result of better (but also mutagenic) cancer treatments and the increased survival of cancer patients [2].

Therapy-related AML accounts for 10–20% of all cases of newly diagnosed AML. Any age group can be affected. Most cases of t-AML associated with a prior neoplastic disorder (~70%) occur after the treatment of solid tumors (e.g., after treatment for breast cancer) and ~30% after treatment of hematological neoplasms, e.g., non-Hodgkin’s lymphoma (NHL) [3,4]. For example, AML can be expected to develop in 2–10% of patients who receive alkylating agents as part of their therapy for breast cancer, ovarian cancer, or NHL [5]. Apart from neoplastic disorders, exposure to cytotoxic agents also occurs during treatment for non-neoplastic disorders. This category includes patients with various autoimmune and inflammatory conditions, such as multiple sclerosis, systemic lupus erythematosus, vasculitis (e.g., Wegener’s granulomatosis and Churg–Strauss syndrome), rheumatoid arthritis, and inflammatory bowel disease, who have been exposed to chemotherapeutic agents (alkylating agents or antimetabolites) as a component of their treatment. In rare cases, t-AML may follow high-dose chemotherapy in the context of treatment of a non-myeloid disorder, e.g., autologous stem-cell transplantation for Crohn’s disease or multiple sclerosis. AML occurring after treatment for non-neoplastic disorders accounts for 5–20% of all cases of t-AML [2].

Therapy-related AML is thought to be the consequence of mutational events or chromosomal changes in primitive hematopoietic stem cells (HSCs) induced by leukemogenic cytotoxic therapy or radiotherapy. It is usually, but not always, associated with cytogenetic lesions [6,7,8,9].

Overall, the prognosis for patients with t-AML is considerably worse than that for patients with primary de novo AML. It is estimated that the median overall survival (OS) is 8–10 months and the 5-year OS is 10–20% [4,10,11,12]. Therefore, allogeneic hematopoietic-cell transplantation (HCT) from a suitable donor has been established practice for all patients with non-acute-promyelocytic-leukemia (non-APL) t-AML in first complete remission (CR1) for more than 30 years [13]. More recently, however, this generalized approach has become controversial due to the increasing recognition of the prognostic significance of the molecular tumor genetics in relation to the risk of relapse [14]. There are clearly no universal right approaches and there may well be different levels of benefit from HCT in the different risk groups of patients with t-AML. Therefore, efforts have been made to individualize treatment even for patients with t-AML. In this context, t-AML is no longer considered separately (with regard to treatment) in the latest European LeukemiaNet recommendations (2022 ELN) [15]. In ELN guidelines, the term “therapy-related” is applied as a qualifier (“diagnostic qualifier”) in the case of AML classified based on its genetic profile, to indicate a history of exposure to cytotoxic agents.

## 2. Etiology of t-AML

The occurrence of t-AML is typically a late adverse effect after the administration of a leukemogenic agent. The main causative agents that have been linked to the development of t-AML include alkylating agents and topoisomerase II inhibitors. Other factors causally associated with t-AML include antimetabolites, anthracyclines (through their ability to inhibit DNA topoisomerases I and II), antimicrotubule (antitubulin) agents (usually in combination with other cytotoxic agents), and ionizing radiation [16]. Moreover, recent studies have reported an increased incidence of MDS and AML after the administration of targeted treatment with the poly (adenosine diphosphate [ADP]–ribose) polymerase (PARP) inhibitors olaparib, niraparib, and rucaparib for metastatic or recurrent epithelial ovarian cancer, particularly in women with germline mutations in *BRCA1* and *BRCA2* (*mBRCA*) genes [17]. The cytotoxic factors associated with t-AML are shown in Table 1.

### Types of t-AML

The most common type of t-AML occurs 4–10 years after exposure to alkylating agents and/or ionizing radiation. The mechanism of action of alkylating agents depends on the creation of bonds, through alkylation, in one or both strands of the DNA double helix [18]. The risk of this complication peaks 5–10 years after the start of chemotherapy. These patients frequently present with MDS, which may then progress to frank AML [19]. This subtype of t-AML is often associated with a loss of genetic material, typically deletions of chromosomes 5 and 7. It is also commonly associated with chromosome 17 or 17p deletion, complex karyotype, and *TP53* mutations [8,16].

A second distinct subtype of t-AML accounting for 20–30% of t-AML has been identified as a complication of treatment with topoisomerase II inhibitors (also called “topoisomerase II poisons”), such as epipodophyllotoxins. In contrast to alkylating-agent-related AML, topoisomerase-inhibitor-related AML develops after a relatively short latency period (1–5 years) and is not preceded by MDS [20]. Epipodophyllotoxins block cells in the late S to G2 phase of the cell cycle. Their major target is the enzyme DNA topoisomerase IIA, a nuclear enzyme that is essential in DNA replication by creating double-stranded cuts in DNA. The binding of epipodophyllotoxins to the enzyme–DNA complex results in persistence of the transient, cleavable form of the complex and, thus, renders it susceptible to irreversible double-strand breaks [21]. Exposure to drugs that inhibit topoisomerase II―i.e., the epipodophyllotoxins etoposide and teniposide; the anthracyclines daunorubicin, doxorubicin, and epirubicin; and the anthracenedione mitoxantrone―predisposes patients to the development of t-AML with balanced chromosomal translocations, including *KMT2A* (*MLL*) translocations at chromosome band 11q23, t(8;21), t(16;16), t(15;17), and t(9;22), and *NUP98* translocations at chromosome band 11p15.5 [15,22].

## 3. Pathophysiology of t-AML

### 3.1. Genetic Predisposition for t-AML

Although many patients are exposed to mutagenic agents as part of their treatment, only a small minority of them develop t-AML in their lifetime. Additionally, among individuals exposed to the same amount of cytotoxic therapy, only a few develop t-AML. This suggests that genetic predisposition may be a key factor.

Clinicians should be aware that as many as 20% of otherwise typical cases of t-AML occurring after treatment for breast or ovarian cancer may actually be AML with germline predisposition caused by inherited mutations in the DNA repair genes (*BRCA1*, *BRCA2*, *PALB2*, *TP53*, or *CHEK2*), typical of familial cancer predisposition syndromes [23,24,25,26]. Somatic (acquired) mutations in *TP53* have been detected in most human cancers including breast cancer. However, inherited mutations (transmitted through the germline) of *TP53* also underlie the Li–Fraumeni syndrome, a rare familial association of breast cancer in young women, leukemia (AML/MDS), childhood sarcomas (“BLS” syndrome), and/or other neoplasms, which is transmitted as an autosomal dominant trait. For example, a woman who has survived breast carcinoma and/or sarcoma and is now facing t-AML with del(17) on karyotype and *TP53* mutation on next-generation sequencing (NGS) should be screened for Li–Fraumeni syndrome (even if her family history is not indicative of it) [22]. Knowing that an important subgroup of younger patients with t-AML carry germline mutations in cancer predisposition genes is important not only for their treatment (e.g., selection of appropriate family donors for allogeneic HCT) but also for identifying family members who may be at high risk for the development of tumors.

Studies have also examined whether polymorphisms in genes involved in the metabolism of alkylating agents and topoisomerase II inhibitors such as cytochrome P450 enzymes (CYP3A4/CYP3A5), GSTM1, GSTT1, and NQO1 may confer an increased risk of t-AML, with so far controversial results [10,27].

### 3.2. Current Model for the Molecular Pathogenesis of t-AML

Beyond genetic predisposition and the inheritance of familial cancer genes, NGS has revealed that, in many patients, the first step in the process of t-AML development is age-related clonal hematopoiesis also known as clonal hematopoiesis of indeterminate potential (CHIP) [28]. According to this model, HSC clones harboring somatic *TP53* or *PPM1D* mutations are detected in patients before chemotherapy exposure [29,30]. Owing to its competitive advantage, the *TP53*-mutant clone enlarges in the bone marrow after chemotherapy administered for a neoplastic or non-neoplastic disorder. The acquisition of additional mutations or the emergence of cytogenetic abnormalities such as chromosome 5/5q or 7/7q deletion leads to a selection of subclones of hematopoietic cells with an increasingly impaired differentiation capacity, which drives leukemic transformation (Figure 1) [8,22,31].

The mutational burden in t-AML is similar to de novo AML, but the relative frequency of specific mutations differs significantly, e.g., mutations in the gene encoding nucleophosmin (*NPM1*) are not as common in t-AML as in de novo AML (Figure 2) [16,32,33]. Notably, *TP53* mutations are the most common molecular abnormality in t-AML.

### 3.3. TP53 Mutations

The *TP53* tumor-suppressor gene, located on the short arm of chromosome 17, encodes a 53-kd nuclear phosphoprotein that suppresses cell growth in response to DNA damage through several mechanisms including cell-cycle arrest at the G1/S checkpoint, activation of DNA-repair enzymes, and initiation of apoptosis in cases of severe DNA damage [9]. From a mechanistic point of view, the activation of the DNA damage checkpoint results in the formation of *TP53* homo-tetramers (tetramerization is essential for p53 activation in vivo). Activated *TP53* then interacts with other tumor suppressors such as p21^Waf1/Cip1^, p63, and p73, activates numerous microRNAs (including the miR-34 family of miRNAs) and proapoptotic proteins (BCL-XL, BCL2, Bax), and alters the function of mTOR kinase.

The transcriptional activity of *TP53* is carried out by five distinct domains including the transactivation activation domain (TAD) and the proline-rich domain (PRD), located at the N-terminal portion of the p53 protein, the core DNA-binding domain (DBD) and the tetramerization domain (TET) (responsible for the oligomerization of the p53 protein, which exists as a tetramer), and the carboxy-terminal regulatory domain (CTD) at the C-terminal portion of the protein. The C-terminal portion also includes several nuclear localization sequences (NLS) (Figure 3) [34,35].

*TP53* is mutated in a large proportion of tumors [34,36]. In fact, acquired *TP53* mutations represent the most common specific genetic change in human cancer. The types of mutations affecting *TP53* include gain-of-function, loss-of-function, and separation-of-function mutations [37,38]. Approximately 18,000 different mutations have been found in different types of malignancies. It is noteworthy that *TP53* shows a specific set of mutations depending on the type of malignancy [34,39]. Mutations that deactivate p53 (loss-of-function mutations) usually occur in the DBD. Most of these mutations impair the ability of the protein to bind to its target DNA sequences, thus preventing the transcriptional activation of p53 target genes. The loss of *TP53* function confers a clonal advantage [9].

In t-AML, the usual mutation (>80%) is a monoallelic missense mutation (i.e., a point mutation in which a single nucleotide is substituted by another, leading to the replacement of a single amino acid) in the DNA-binding domain [40,41,42]. The result of this mutation is the abnormally increased expression of the mutant p53 protein due to a longer half-life compared to the wild-type protein (in normal tissues, p53 protein is present in very low quantities so that it is not readily detectable by immunochemistry. However, in *TP53*-mutated leukemic cells, large amounts of p53 protein accumulate in the nucleus which can be seen by staining [“p53 over-expression”]; this is generally attributed to the accumulation of over-stabilized, mutant protein). Less frequent mutations include deletions, truncations, insertions, nonsense, and splice-site mutations [40,41,43]. The definition of t-AML with *TP53* mutation requires the presence of a somatic *TP53* mutation with a variant-allele frequency [VAF] of >10%. Monoallelic *TP53*-mutated t-AML has a poor prognosis. The term “multi-hit” mutations refers to the presence of two distinct *TP53* mutations (each with a VAF of >10%) or a single *TP53* mutation with either (i) 17/17p deletion on cytogenetics; (ii) copy-neutral loss of heterozygosity (LOH) at the 17p (*TP53* locus); or (iii) VAF of >50%, which is 75% concordant with copy-neutral LOH variants [44,45].

Different *TP53* mutations occur at different stages in the course of AML. For example, biallelic defects and multi-hit events take place at a very early stage during leukemogenesis. These mutations contribute to the creation of a dominant clone [46,47]. Monoallelic *TP53* mutations, on the other hand, usually occur at later stages of the disease as subclonal events [46,47]. Monoallelic *TP53* mutations may also coexist with other driver mutations such as *SF3B1*, *TET2*, *DMT3A*, and *ASXL1* mutations [46,47]. *TP53* mutations are associated with complex or monosomal karyotypes and “chromosome shattering” (also known as “chromothripsis”), a phenomenon characterized by extensive chromosomal rearrangements [48,49,50].

An important development is the discovery that p53 activates the transcription of a set of microRNAs, including the miR-34 family. Pathogenic mutations in *TP53* result in diminished expression of microRNA-34a (miR-34a), a potent tumor-suppressive microRNA, leading to the over-expression of *c-MYC* oncogene and upregulation of PD-L1 in tumor cells [22,51].

### 3.4. Molecular Basis of t-AML

Lindsley and coworkers investigated the genetic basis of t-AML and secondary AML (s-AML) and identified three distinct subtypes of somatic mutations: (i) secondary-type mutations involving eight genes (*SRSF2*, *SF3B1*, *U2AF1*, *ZRSR2*, *ASXL1*, *EZH2*, *BCOR*, and *STAG2*) which are commonly linked to MDS and are now recognized as “AML with myelodysplasia-related (MR) gene mutation” in 2022 ELN recommendations; (ii) *TP53* mutations, which are associated with complex karyotypes (often monosomal, with frequent abnormalities of chromosomes 5 and 7), intrinsic therapy resistance, and very poor survival; and (iii) “de novo”-type (or pan-AML-type) mutations including *NPM1* mutations, *KMT2A (MLL)* rearrangements at 11q23 locus, core-binding-factor (CBF) chromosomal rearrangements, myeloid-transcription-factor mutations (e.g., *RUNX1*, *CEBPA*, and *GATA2*), signal-transduction-protein mutations (e.g., *FLT3*, *N-RAS*, and *K-RAS*), and other mutations (e.g., *IDH1, IDH2*, and *WT1*) [8].

“Secondary-type” mutations are found in 30% of t-AML cases and are associated with poor outcomes. *TP53* mutations are seen in ~50% of t-AML cases. *NPM1* mutations are identified in 5.4% of patients without concurrent “secondary-type” or *TP53* mutations [8,30,52], suggesting that there is a non-random pattern of co-mutations with mutual exclusivity between *TP53* and *NPM1* mutations in t-AML. Patients with t-AML with “secondary-type” mutations are significantly older than patients with “de novo”/pan-AML mutations (Figure 4). Clinically, t-AML with “secondary-type” mutations closely resembles s-AML. In contrast, patients with t-AML with “de novo”-type mutations closely resemble patients with primary de novo AML [8,53].

Recently, Papaemmanuil and coworkers presented another categorization of patients with AML into 16 distinct molecular subgroups [54]. Regarding t-AML, the majority of patients had high relapse rates and poor prognosis, regardless of the achievement of early minimal residual disease (MRD) negativity. However, beyond this general rule, patients with t-AML who achieved CR, including those with *TP53* mutations, seemed to benefit from HCT. Additionally, patients who had >2 mutations had worse prognosis compared to those who carried a single-gene mutation. The prognosis was even worse in patients with *TP53* mutations or inv(3)/t(3;3), resulting in deregulated *MECOM* (*EVI1*) and *GATA2* expression [54]. However, it should be emphasized that even patients carrying *TP53* mutations benefited from HCT, especially in CR1 (and less so after CR2) [54].

Clearly, the past decade has reshaped our view of t-AML. Rather than considering t-AML as one clinical entity, it is more appropriate to view t-AML as at least three molecular types (according to the Lindsley model) that vary in prevalence with age, each bearing more similarity to AML with the same genetic alterations and no leukemogenic exposure.

### 3.5. A Permissive Bone Marrow Microenvironment Facilitates t-AML Growth

The immune system inside the bone marrow is also involved in the development of t-AML. Population-based studies show that AML is more common among patients with autoimmune diseases than the general population [55,56]. The risk of developing t-AML appears to be related to the type of autoimmune disease and type and duration of its treatment. Drugs used to treat autoimmune diseases such as azathioprine, mitoxantrone, and cyclophosphamide may directly damage DNA and increase the risk of leukemogenesis [55,56]. However, drugs used in autoimmune disorders such as azathioprine may also affect the balance between T-cell subsets. Experiments in mice have shown that high doses of azathioprine downregulate regulatory T-cells (T-regs), whereas lower doses upregulate T-regs [57]. Abnormalities in T-regs may play a role in the transition of MDS to AML [58,59]. For example, in patients with clinically stable MDS (regardless of the disease stage), T-reg levels remain stable, but upon transformation to AML, an increase in T-reg numbers is noted both in the marrow and in the peripheral blood [60]. In addition, reduced immune surveillance by cytotoxic T-cells is seen in patients with autoimmune disorders. Chronic inflammatory signaling and inflammatory conditions can modulate the bone marrow microenvironment and facilitate the survival and proliferation of leukemic cells [56,61]. Notably, the transcription factor nuclear factor-κB (NF-κB), a central pro-inflammatory mediator, and polymorphisms in the interleukin-1 receptor antagonist (IL-1Rα) are involved in both autoimmune diseases and leukemogenesis [56,62,63,64,65].

### 3.6. Role of Pro-Inflammatory Cytokine Signaling

The response of a body to a cancer is not a unique mechanism but has many parallels with the chronic inflammation seen in chronic infections. Balkwill and Mantovani’s metaphor, stating that if genetic damage is “the match that lights the fire of cancer”, some types of inflammation provide “the fuel that feeds the flames”, puts emphasis on inflammation as a major contributor in the growth of some cancers [66]. Multifaceted activation of the immune system and chronic inflammation accompany many hematologic neoplasms including MDS and AML. Certain pro-inflammatory cytokines such as the tumor necrosis factor α (TNF-α), interleukin-1 (IL-1), and interleukin-6 (IL-6) play an important role in the development of AML. IL-1 occurs as two structurally related polypeptides (IL-1α and IL-1β), each of which has a broad spectrum of both beneficial and harmful biologic effects. IL-1β, in particular, contributes to the proliferation and survival of leukemic cells [65,67]. It has been found that IL-1β participates in the immune response through a dual mechanism: the activation of the IL-1 receptor (IL-1R)/Toll-like receptor (TLR) and caspase-1 activation [68,69]. The activation of IL-1β induces a signaling cascade that leads to the phosphorylation and ubiquitination of MyD88, IRAK-4, and TRAF-6 and, ultimately, NF-κB activation [70]. At the same time, IL-1β induces the activation of p38 mitogen-activated protein kinase (MAPK), as well as the activation of the transcription factor *GATA2* [70,71,72,73]. Both p38 MAPK and *GATA2* contribute to the proliferation of leukemic blasts. Monocytes and macrophages represent the main cell source of IL-1β. IL-1β may be a target for future therapies in AML.

IL-35, transforming growth factor β (TGF-β), and IL-10 have also been implicated in the pathogenesis of AML. IL-35 belongs to the IL-12 family and is produced by T-regs. In patients with AML, high concentrations of IL-35 have been found in bone marrow plasma, corresponding to an increased proportion of T-regs [74,75,76]. TGF-β contributes to many cellular processes including cell survival, proliferation, and migration. In AML, TGF-β inhibits the proliferation of leukemic stem cells (LSCs), maintaining their longevity.

The administration of DNA damaging agents, in association with disturbances in T-cell subsets and pro-inflammatory changes in the bone marrow microenvironment, may favor the development of t-AML in patients with autoimmune disorders.

## 4. Management

### 4.1. Individualized Risk Assessments in t-AML

The risk assessment of patients with t-AML should be individualized based on the 2022 ELN genetic risk classification (Table 2) [15]. According to ELN recommendations, “genetic aberrations are given priority in AML classification, with additional predisposing features (e.g., therapy-related AML, secondary AML, or germline predisposition) appended as qualifiers of the primary diagnosis” [8,64]. Thus, t-AML should be investigated and classified like primary de novo AML. For example, in patients with *NPM1*-mutated t-AML, it is crucial to exclude karyotypic abnormalities and *FLT3* internal tandem duplication *(FLT3*-ITD). Normal-karyotype *NPM1*-mutated t-AML without *FLT3*-ITD is classified as a favorable-risk disease. In contrast, *NPM1*-mutated t-AML with adverse cytogenetics such as -7 or complex karyotype―as often occurs in t-AML―is categorized as adverse-risk disease by ELN. Currently, the impact of intermediate-risk cytogenetic lesions, e.g., +8 or +4, in patients with *NPM1*-mutated t-AML remains unclear [77].

### 4.2. Overall Prognosis of t-AML

Historically, patients with t-AML have been under-represented within clinical trials and excluded from large-scale studies. Excluding the rare patient with t-AML who belongs to the 2022 ELN favorable-risk group, the overall prognosis of t-AML remains very poor [1,9,22,78]. This is not only because of the high frequency of adverse cytogenetic (e.g., -7/-7q, -5/-5q, -3/-3q, -17/-17p, complex and/or monosomal karyotypes) and molecular lesions (e.g., *TP53* mutations), but also due to the sequelae of prior chemotherapy and sometimes persistent primary disease, particularly metastatic cancer or lymphoma.

### 4.3. Is t-AML with Favorable Genetic Lesions as Favorable as De Novo AML?

There have been conflicting reports regarding the incidence and prognosis of t-AML with favorable-risk lesions. In 2004, Schoch and colleagues showed that the prognosis of patients with t-AML was more favorable than s-AML, and that patients with t-AML and intermediate or unfavorable cytogenetics had similar OS with their de novo counterparts, whereas age and white-cell count had no impact on OS in t-AML [79]. However, this was not the case for patients with t-AML with CBF rearrangements (i.e., t(8;21) or inv(16)/t(16;16)); these patients appeared to have inferior OS compared to de novo CBF AML [79]. In comparing t-AML and de novo AML with favorable cytogenetics, Aldoss and Pullarkat concluded that a history of prior cytotoxic treatment can affect outcome in CBF AML but not to the degree of advocating a change in treatment strategy or transplantation indications for CBF t-AML. In addition, they showed that OS rates were similar between patients with de novo and therapy-related APL [80].

Gemtuzumab ozogamicin (GO), a humanized anti-CD33 monoclonal antibody conjugated with the cytotoxic agent calicheamicin, has shown positive results in newly diagnosed CBF AML. A meta-analysis of five randomized trials confirmed that although adding GO to standard chemotherapy did not increase response rates, it reduced the risk of relapse and significantly improved OS among younger and older adults with CBF AML [81,82]. Thus, GO might be expected to benefit patients with CBF t-AML, too.

Falini and colleagues first demonstrated that the abnormal cytoplasmic localization of nucleophosmin resulted in increased responsiveness to induction chemotherapy in patients with normal-karyotype AML [83]. In 2008, a Danish study tried to evaluate the frequency of *NPM1* mutations in t-AML. Among 140 patients with t-AML, they identified 10 patients harboring *NPM1* mutations, of whom four had a normal karyotype and raised the question whether these four patients actually represented “de novo AML with medical history” rather that true t-AML [84,85]. Another Danish study, however, confirmed that *NPM1*-mutated t-AML without karyotypic lesions does exist [86]. In a Swedish national cohort study of 6779 patients with AML including 686 with t-AML, Nilsson and coworkers showed that after adjusting for age and performance status, survival was similar in favorable-risk therapy-related and de novo AML (hazard ratio [HR] 0.99; 95% confidence interval [CI] 0.89–1.11; *p* = 0.95). They concluded that the best combination for survival was *NPM1*-mutated/*FLT3* wild-type, and that such patients could be approached therapeutically similarly to patients with de novo AML. Notably, the presence of *FLT3*-ITD did not seem to have a negative impact on survival in t-AML (perhaps due to their low frequency), while *NPM1*-mutated t-AML carried significantly better survival rates than *NPM1* wild-type t-AML. This study also revealed that patients with t-AML were less likely to receive intensive chemotherapy and undergo HCT than their de novo counterparts. The study by Nilsson and colleagues, unlike the study by Schoch, demonstrated worse OS in t-AML with intermediate-risk and adverse cytogenetics and similar OS in t-AML with favorable-risk cytogenetics, compared to de novo AML [79,87].

### 4.4. Therapeutic Approach of Patients with t-AML

As a rule, patients with t-AML should be managed according to the same general therapeutic principles (i.e., according to whether they are candidates for intensive or non-intensive therapy and allogeneic HCT) as their de novo counterparts.

In practice, the patient’s age, general health/performance status, nutritional status, specific coexisting conditions/comorbidities, possible complications from the original disease or previous treatments, sequelae of prior chemotherapy (e.g., cardiac injury as a consequence of anthracycline exposure or cisplatin-induced chronic kidney disease), cumulative dose of anthracyclines, chronic immunosuppression from prior disease or ongoing therapy, history of severe infections, and status and prognosis of the underlying primary malignant or autoimmune disorder, as well as the disease features, the patient’s wishes (and those of the patient’s relatives, including support at home) and emotional status, and the physician’s attitude and interest all influence clinical decision making [88]. It is important to note that there may be a depletion of normal HSCs, and the bone marrow stroma may have been damaged as a consequence of previous chemotherapy or radiotherapy, so patients with t-AML may suffer prolonged cytopenias after additional chemotherapy. Following prior supportive care, patients may be refractory to additional transfusion support, and therefore not good candidates for myelosuppressive chemotherapy [88].

Patients considered fit for intensive chemotherapy should be treated with standard anthracycline plus cytarabine induction chemotherapy (“3 + 7”), followed by standard consolidation and/or allogeneic HCT, depending on their 2022 ELN genetic risk (see Figure 5 for an example). CPX-351, a liposomal formulation of cytarabine and daunorubicin at a fixed 5:1 molar ratio, represents an alternative intensive (induction) chemotherapy option for these patients. In fact, CPX-351 is the first approved treatment specifically for patients with t-AML. Its approval was based on findings from a multicenter, randomized, open-label, phase III study of CPX-351 versus “3 + 7” in patients 60–75 years old with newly diagnosed t-AML. CPX-351 had higher CR and CR with incomplete count recovery (CRi) (47.7% versus 33.3%), and higher median OS (9.56 versus 5.95 months) rates than “3 + 7” chemotherapy. The 5-year OS with CPX-351 was 18% versus 10% for those receiving “3 + 7”. Notably, CPX-351 delayed the median time to neutrophil and platelet count recovery by approximately seven days and increased the risk of bleeding. Early 30-day mortality, however, was not increased (5.9% versus 10.6%) [89,90,91]. No randomized data exist for the use of CPX-351 in younger patients (age <60 years) with t-AML.

Patients not considered medically fit for intensive chemotherapy (i.e., older or frail patients) are usually treated―outside of the context of a clinical trial―with the hypomethylating agents (HMA) azacitidine or decitabine, either alone or in combination with venetoclax, an oral selective inhibitor of the anti-apoptotic protein BCL-2.

### 4.5. Can Mutations Predict Response to HMA-Venetoclax Combination?

It has been shown that patients with AML harboring mutations in *NPM1*, *IDH1/2*, or *TET2* (and possibly *RUNX1* and *STAG2* in the relapsed/refractory setting) respond favorably to azacitidine–venetoclax (or decitabine–venetoclax), independently of the underlying karyotype. In particular, the 1-year survival rate of *NPM1*-mutated AML receiving venetoclax with azacitidine was 80% versus only 12% for azacitidine alone and 56% for induction chemotherapy [92]. Similarly, the azacitidine–venetoclax combination achieved a higher CR/CRi rate than induction chemotherapy (56–89% versus 61%) in newly diagnosed *IDH*-mutated AML, particularly *IDH2*-mutated AML [93,94]. *TET2* mutations are also very sensitive to azacitidine–venetoclax, with a CR/CRi rate of ~86% (versus 39% for *TET2* wild-type AML) [93,94]. Patients with relapsed/refractory *RUNX1*-mutated AML may also respond favorably to the venetoclax with azacitidine or decitabine with an expected CR/CRi rate of 35–75% and prolonged OS. However, newly diagnosed *RUNX1*-mutated AML is less sensitive to azacitidine–venetoclax than *RUNX1* wild-type AML (CR/CRi rate 50% versus 64%) [94,95]. In a recent study of 86 patients treated at the Memorial Sloan Kettering Cancer Center (New York, NY, USA), a novel association between *STAG2* mutations and improved survival after azacitidine–venetoclax combination was reported, albeit only when azacitidine–venetoclax was administered for relapsed/refractory AML [96].

In contrast, *TP53*, *PTPN11*, *N-RAS*/*K-RAS*, *U2AF1*, and perhaps *FLT3-ITD* mutations have been associated with poor outcomes and, possibly, reduced sensitivity to azacitidine–venetoclax. The co-occurrence of high-allelic-ratio *FLT3*-ITD with *NPM1* mutations adversely affected outcomes compared to *NPM1* mutations alone or *NPM1* mutations with low-allelic-ratio *FLT3*-ITD [93]. The median OS of *FTL3*-mutated AML treated with intensive chemotherapy was 5.8–26 months versus 12.5 months for azacitidine–venetoclax and 8.6 months for azacitidine alone [97]. *N-RAS*, *K-RAS*, and *PTPN11* mutations are prevalent in t-AML. When treated with HMA–venetoclax, *N-RAS* mutations are associated with poor response rates (CR/CRi 0–36%) and shorter median OS (3.8 months), compared with patients with wild-type *N-RAS* (7.4 months) [93,94,95]. Patients with *TP53* mutations respond poorly and relapse early after conventional chemotherapy (2-year OS 12.8–14%) [98]; in addition, they respond poorly to azacitidine–venetoclax (median OS 1.9 months for relapsed/refractory *TP53*-mutated AML) [95].

Spliceosome-complex mutations (*SRSF2*, *ZRSR2*, *U2AF1*, and *SF3B1*) have variable sensitivity to combination therapy, possibly due to the co-occurrence of other mutations. For example, *SRSF2* mutations, which are often associated with *IDH2* mutations, have demonstrated favorable response to azacitidine–venetoclax, with a 1-year and 2-year OS of 100% and 87%, respectively [99]. Notably, *IDH2* mutations preferentially co-occur with *SRSF2* mutations rather than other mutations in RNA splicing machinery (88% versus 11%). By contrast, *U2AF1* mutations co-occur with *RAS* mutations, suggesting that the resistance and early relapse of *U2AF1*-mutated AML after azacitidine–venetoclax may be the result of *RAS* co-mutation rather than of *U2AF1* itself [93,100]. *SF3B1*-mutated AML has been identified as a marker of venetoclax resistance [101], and it remains to be seen whether HMA–venetoclax, especially decitabine–venetoclax (since *SF3B1*-mutated AML appears to be particularly sensitive to decitabine-based therapy) [102], can overcome resistance through a synergistic effect [103].

In vitro studies have shown that *ASXL1* mutations are associated with BCL-2 overexpression and increased levels of global cytosine hypermethylation, suggesting sensitivity to venetoclax–azacitidine [104]; however, this hypothesis requires prospective confirmation.

In one study, *DNMT3A*-mutated relapsed/refractory patients who were naive to HMAs achieved high response rates and prolonged survival with venetoclax combination therapy, whereas *DNMT3A*-mutated patients who had previously received HMAs responded poorly to HMA–venetoclax, with poor survival outcomes [96]. This may suggest sensitivity of *DNMT3A* mutations to epigenetic modification. Many studies to date have focused on the metabolic and apoptotic effects of these combinations (e.g., reduced mitochondrial oxidative phosphorylation, lower expression of amino-acid transporters, and increased release of cytochrome c [cyt c] in LSCs), but it is also important to investigate the mechanism by which venetoclax may act synergistically with HMAs to modify epigenetic targets in *DNMT3A*-mutated AML. It is not expected that clonal hematopoiesis (particularly *DNMT3A* mutations) will be eradicated after first-line HMA-based treatment.

Despite their widespread use, it is important to note that HMAs have not significantly improved the outcome of patients with t-AML; the median OS of t-AML treated with first-line HMA is approximately 7 months. A recent study confirmed the poor outcome of t-AML treated with HMAs [105]. The outcome was particularly poor in t-AML with *TP53* mutations and/or complex karyotype. In the phase III randomized control trial VIALE-A, the addition of venetoclax to azacitidine for patients with AML who were ineligible for “3 + 7” because of age (≥75 years), coexisting conditions, or both led to remarkable improvements in CR/CRi rates (66.4% versus 28.3%) and median OS (14.7 months versus 9.6 months) compared to azacitidine alone [106]. This study included 62 patients with t-AML (36 in the azacitidine–venetoclax arm and 26 in the azacitidine–placebo [control] arm). Azacitidine added to venetoclax led to only modest improvement in patients with *TP53* mutations or complex karyotype (CR/CRi, ~20%; median OS <12 months). A phase II study found that a 10-day course of decitabine improved the outcome of patients with *TP53*-mutated AML enough to resemble that of intermediate-risk AML [64,107]. However, this finding requires prospective confirmation with a larger study. Another study analyzed 378 patients with therapy-related myeloid neoplasms, of which 96 (25.4%) received venetoclax (47 t-AML, 48 t-MDS, and one patient with t-MDS/MPN) [108]. Despite initial response, early progression was seen in most cases as a result of adverse disease biology (mutations in signaling kinases and biallelic loss of *TP53*, leading to venetoclax resistance). Clearly, there is an urgent need for new treatments for these patients.

## 5. Guidelines from Professional Societies

For patients with t-AML aged ≤60 years who are deemed fit for intensive chemotherapy, National Comprehensive Cancer Network (NCCN) guidelines recommend either standard induction chemotherapy (“3 + 7”) or CPX-351, whereas for patients aged ≥60 years who are eligible for chemotherapy, NCCN guidelines recommend CPX-351 [109]. Similarly, European Society for Medical Oncology (ESMO) guidelines recommend treatment with CPX-351 for patients with t-AML who are eligible for standard intensive chemotherapy and HMA–venetoclax combinations for those who are ineligible for intensive chemotherapy [110].

## 6. Authors’ Recommendations for Individualizing Treatment in t-AML

Table 3 shows examples of biomarker-driven, individualized treatment approaches for patients with t-AML.

## 7. Emerging Treatments

Although CPX-351 offers something new in the landscape of t-AML therapy, most patients with t-AML are less chemosensitive than those with primary AML and, therefore, do not benefit from either CPX-351 or “3 + 7”. The efforts made by the researchers over the previous decades to improve chemotherapy regimens have not led to major improvements, until the recent evolution seen with new molecularly targeted therapies directed at specific mutations. Ongoing research and clinical trials are actively seeking ways to personalize therapy in t-AML by identifying new targets, discovering patient-specific and disease-specific risk factors, and finding effective combinations of new agents targeting multiple cellular processes (Table 4).

Magrolimab (Hu5F9-G4) is a first-in-class humanized IgG4 monoclonal antibody against CD47 and prompts cancer cell phagocytosis by macrophages through the disruption of the CD47–signal-regulatory-protein-alpha (SIRPα) inhibitory checkpoint, thereby blocking the “don’t eat me signal”. CD47 is also a marker of LSC, and targeting CD47 can potentially eliminate LSC while sparing normal HSC [111]. Magrolimab showed synergism with azacitidine in preclinical studies using AML cell lines, so this combination was explored in a phase Ib trial that included older/unfit patients with higher-risk MDS and newly diagnosed AML who were ineligible for induction therapy. Azacitidine with magrolimab demonstrated an overall response rate (ORR) of 49% and a CR rate of 33% among older/unfit patients with *TP53*-mutated AML treated in this trial. The median duration of response was 8.7 months and the median OS was 10.8 months. In another study, patients with newly diagnosed *TP53*-mutated AML (n = 14) were treated with magrolimab in combination with venetoclax and azacitidine. The ORR was 86%, with a CR rate of 64%, MRD-negativity of 55%, and substantial clearance of *TP53*-mutated clones in eight out of nine patients who achieved CR/CRi [112].

Sabatolimab is a humanized, high-affinity IgG4 monoclonal antibody targeting TIM3, a myeloid checkpoint that forms part of a co-inhibitory receptor complex overexpressed on exhausted T-cells and also on LSCs in patients with AML/MDS. A phase Ib trial evaluated sabatolimab in combination with HMAs in newly diagnosed patients with AML (n = 48) and higher-risk MDS (n = 53) unfit for intensive chemotherapy. The CR/CRi rate was 40% in patients with *TP53*-mutated AML with a median duration-of-response of 6.4 months [113]. Another molecule that is expressed in about 90% of the AML blasts is CD123, which serves as the receptor for IL-3, and its downstream signaling promotes hematopoietic progenitor-cell proliferation through the activation of the PI3K/MAPK pathway and upregulation of antiapoptotic proteins. CD123 has emerged as a potential target in AML. For example, flotetuzumab, a bispecific antibody that binds CD123 and CD3ε (CD123 × CD3), is under investigation in AML. This molecule promotes T-cell activation and proliferation, resulting in the eradication of CD123-expressing AML blasts. It has been evaluated in a phase I/II study in relapsed/refractory AML, enriched with patients with primary induction failure or early relapse (within 6 months of response) [114]. Among patients with relapsed/refractory *TP53*-mutated AML, the overall response rate was 47% with an encouraging median OS of 10.3 months in responding patients.

### TP53 Pathway Modulation

Given the frequency of *TP53* mutations in t-AML and their poor prognosis, particular reference is required for novel agents that target the *TP53* pathway. The frequency of *TP53* abnormalities increases to 70–80% in patients with complex karyotype, -17/17p, -5/5q, and -7 [115,116]. As mentioned, the high frequency of *TP53* mutations in t-AML may be explained by the observation that pre-existing progenitor cells that carry a *TP53* mutation are resistant to DNA damage and, therefore, can expand under selective pressure from chemotherapy administered for a non-myeloid disorder [9,117].

The presence of *TP53* mutations is inherently problematic for patients receiving cytotoxic chemotherapy since the mutated p53 protein causes dysregulation of the apoptotic pathway, decreasing the efficacy of DNA-damaging chemotherapy agents. *TP53* mutations, especially multi-hit, result in poor clinical outcomes regardless of whether the underlying disorder is classified as MDS or AML on the basis of blast-cell count. Multi-hit t-MDS/AML often represents a distinct stem-cell disorder with a paucity of co-mutations in other myeloid malignancy-related genes (co-mutations occur in <20% of cases). As mentioned, another adverse-risk biological characteristic associated with *TP53* mutation and complex karyotype in t-AML is chromothripsis, which is a catastrophic event leading to extensive chromosomal rearrangement and defines a subset of complex-karyotype AML with an extremely poor outcome [49].

Eprenetapopt (APR-246) is a first-in-class drug that binds covalently to cysteine residues in the DBD domain of the mutant p53 protein. It has been suggested that this binding would induce mutant p53 protein to fold back into its active, wild-type conformation and function. Two studies evaluated eprenetapopt-azacitidine in patients with newly diagnosed MDS, AML, and MDS/MPN; they found a significantly higher CR rate in patients with *TP53* mutations only (52% versus 30% in patients with co-mutations), including those with biallelic *TP53* mutations or complex karyotype (CR 49%) [118].

In search of potential therapies, studies have also evaluated the immune microenvironment and cytokine milieu of *TP53*-mutated AML. *TP53* mutations are associated with an accumulation of immune-inhibitory checkpoints including PD-L1 on LSCs, TIM-3 on myeloid-derived suppressor cells (MDSCs), and LAG3 and TIGIT on leukemic blasts. Furthermore, *TP53*-mutated MDS and AML is characteristically associated with an immunosuppressive marrow environment with FOXP3 overexpression, increased numbers of activated (ICOS^high^) T-regs and PD1^+^ MDSCs, decreased numbers of OX40^+^ cytotoxic T-cells and NK cells, and a marked impairment of CD28^+^ T-cells to secrete immune-effector T-helper-1 cytokines [51,119]. Such profound immune dysregulation, with features of immunosenescence and immune evasion, makes considerable biological sense for the use of novel immunotherapy (antibody-based) approaches such as magrolimab, sabatolimab, and bispecific antibodies in *TP53*-mutated t-AML [120].

## 8. Conclusions

The occurrence of t-AML is an important complication of the treatment of a variety of tumors and autoimmune disorders. More than two thirds of these patients have adverse prognostic features. Over the past decade, an understanding of the molecular pathways leading to the development of AML has resulted in significant improvements in the risk stratification of patients and offered the opportunity to develop individualized therapeutic approaches that target disease biology. The development of new targeted drugs and treatment strategies have improved the outlook of patients with AML, including t-AML. Despite these improvements, however, survival rates for most patients with t-AML remain poor. Clearly, new approaches to therapy are needed in this large cohort of patients. In general, the trend in the treatment of t-AML is toward the modification of therapy to treat specific subtypes of the disease (“biomarker-driven” strategy) and, more specifically, the targeting of the leukemic cells with highly efficacious molecular-targeting agents and/or immunologic (antibody-based) therapeutic strategies [120]. Current clinical investigation is focused on the discovery of new treatments that are intended to provide an improvement in efficacy over existing therapies. Trials are underway to determine whether novel agents can improve cure rates for patients with t-AML. In the meantime, we ought to focus on the optimal use of currently available treatment options to ensure excellent clinical care [121]. It seems appropriate to quote William Osler; “The best preparation for tomorrow is to do today’s work superbly well”.

## Figures and Tables

**Figure 1 cancers-15-01658-f001:**
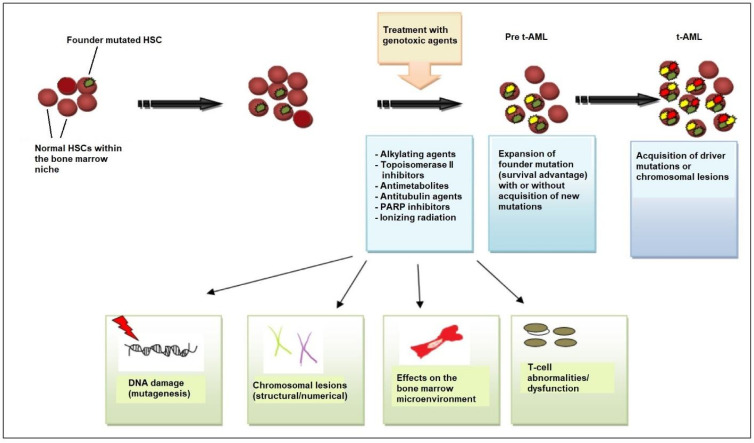
Pathogenesis of t-AML. Evidence supports the view that chemotherapy and/or radiation selects for the expansion of a mutant clone (founder clone) in the hematopoietic-stem-cell (HSC) and/or progenitor-cell compartment of the bone marrow that is more resistant to DNA damage. After chemotherapy/radiotherapy exposure, the pool of normal HSCs is affected and depleted, allowing the mutant clone to expand, owing to its survival advantage. Cytotoxic therapy affects the stem cell, causing DNA damage (i.e., additional mutations and/or chromosomal abnormalities) that drives the emergence of fully leukemic clones and changes in the bone marrow microenvironment, including bone marrow stromal cells and T-cell subsets (cytotoxic T-cells, regulatory T-cells). Stromal cells support HSC function but, if challenged with genotoxic agents, can inhibit normal HSC survival, providing an opportunity for the selection of pre-leukemic clones, and T-cell dysfunction interferes with tumor immunosurveillance. All these mechanisms contribute to the development of leukemic cells and ultimately to the emergence of t-AML. Abbreviations: t-AML, therapy-related acute myeloid leukemia; PARP, poly (ADP-ribose) polymerase.

**Figure 2 cancers-15-01658-f002:**
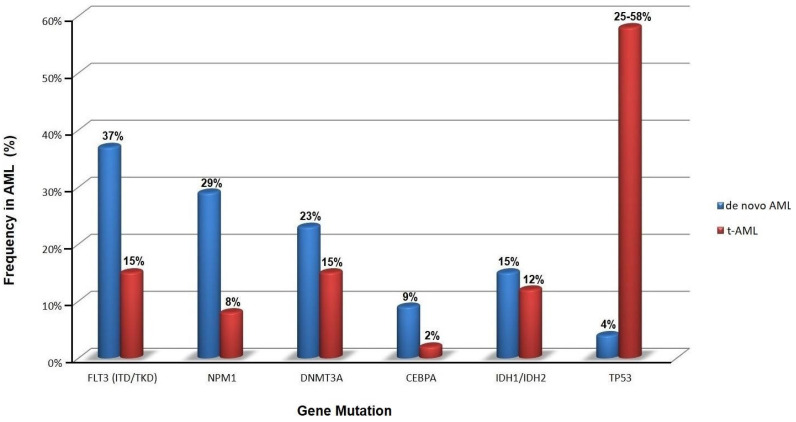
Frequency of somatic mutations in primary (de novo) and therapy-related acute myeloid leukemia (t-AML) at onset (data obtained from [16,32,33]).

**Figure 3 cancers-15-01658-f003:**
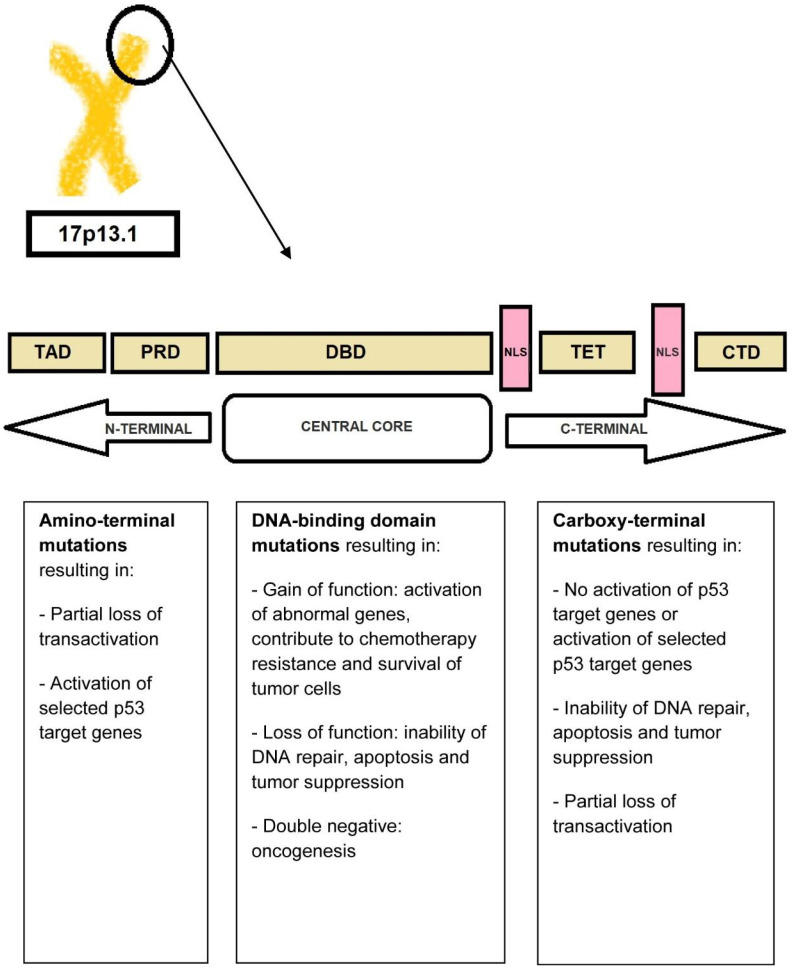
P53 protein structure. The N-terminal portion of the p53 protein includes the transactivation domains (TAD) I and II and the proline-rich domain (PRD). The central core of the protein includes the DNA-binding domain (DBD). The C-terminal portion of p53 includes the tetramerization domain (TET) required for p53 tetramerization and activation, several nuclear localization sequences (NLS), and the carboxy-terminal regulatory domain.

**Figure 4 cancers-15-01658-f004:**
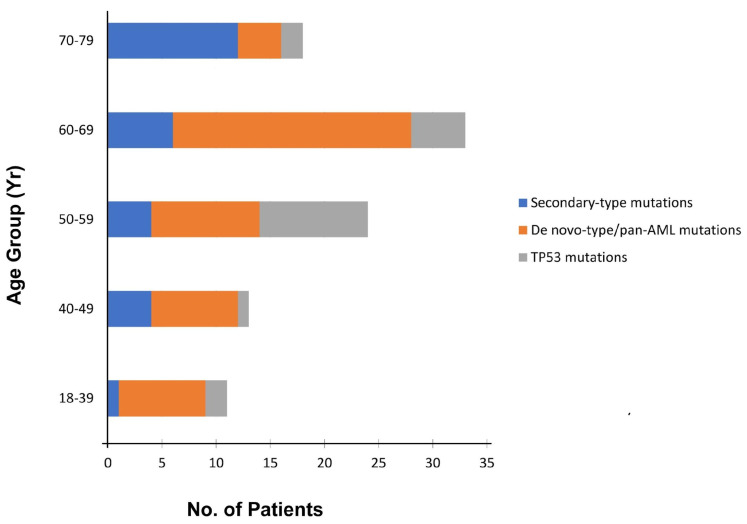
Distribution of the three distinct subtypes of somatic mutations (“secondary-type”, *TP53*, and “de novo”-type/pan-AML mutations) among 217 patients with t-AML, according to age at onset (modified from [8]).

**Figure 5 cancers-15-01658-f005:**
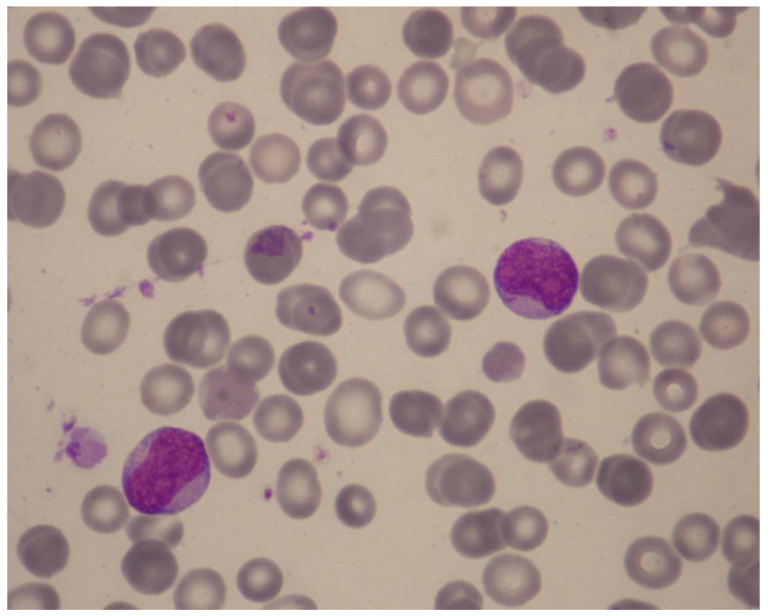
Illustrative case of t-AML. A 49-year-old woman was referred because of neutropenia and thrombocytopenia of 2 months’ duration that were found on routine tests. Four years previously, she had been treated for Churg–Strauss syndrome with steroids and intravenous pulse cyclophosphamide 500 mg once per month for 6 months. Her history included cold-agglutinin disease, osteoporosis, and anxiety with coexisting depressive disorder. Her medications included low-dose prednisolone and azathioprine. The hemoglobin was 14.0 g/dL, white-cell count 9140/μL (neutrophils, 820/μL), MCV 86.6 fL, and platelet count 58,000/μL. May-Grünwald-Giemsa staining of a peripheral-blood smear revealed 58% blasts with Auer rods, easily identifiable under oil immersion (×1000) (Figure). Bone marrow examination demonstrated a moderately cellular marrow with 38% leukemic blasts expressing CD34^partial^, CD117, CD13, CD33, and HLA-DR. A diagnosis of AML without maturation was made. She had a normal karyotype and an exon 12 *NPM1* mutation without *FLT3*-ITD or other lesions on a targeted NGS panel. How should she be treated? Recommendation: The patient has t-AML post cytotoxic/immunosuppressive therapy with an alkylator (cyclophosphamide) and antimetabolite (azathioprine). The type of t-AML induced by cyclophosphamide is usually preceded by MDS which then progresses to overt AML. Our patient’s 2-month history of neutropenia and thrombocytopenia is not sufficient to document preceding t-MDS (according to the minimal diagnostic criteria for MDS, persistent cytopenia lasting for ≥4 months is required for a diagnosis of MDS). Her disease is classified according to ELN as “AML with mutated *NPM1*, therapy-related”. She belongs to the ELN favorable-risk group. Important considerations for definitive therapy in this patient include prognosis of her original non-malignant disease and possible complications from the original disease or previous treatments. Additional work-up showed ejection fraction 55%, oxygen saturation 96%, and slightly diminished lung volumes on a chest CT scan. Therefore, she was deemed fit for intensive chemotherapy. She received “3 + 7” chemotherapy and went into CR. Her clinical course was complicated by neutropenic fever, soft-tissue infection, and platelet-transfusion refractoriness due to platelet HLA alloimmunization. She was found to have a 40-year-old, fully matched sibling donor. ELN suggests HCT in CR1 for fit patients with t-AML who have poor or intermediate risk genetic features. Our patient belongs to the favorable group and therefore we did not refer her for HCT. The treatment plan included a second cycle of anthracycline-cytarabine chemotherapy, followed by three cycles of consolidation with intermediate-dose cytarabine (1.5 g/m^2^), reserving HCT in case of disease relapse. Although there is evidence supporting the use of gemtuzumab ozogamicin in *NPM1*-mutated AML, it was not administered due to concern for excess myelosuppression in this patient who already had platelet refractoriness and a diminished bone marrow reserve stem-cell pool as a result of prior chemotherapy and prolonged azathioprine use. *NPM1* mutations are particularly suitable for assessing MRD since they are typically stable at relapse and do not drive clonal hematopoiesis. Thus, the patient’s *NPM1* mutation can be used as a surrogate marker to assess the probability of relapse. Molecular MRD assessment should be performed at the end of her treatment by means of a real-time qPCR assay, and then monitored every 3 months for 2 years using bone marrow samples. Abbreviations: t-AML, therapy-related acute myeloid leukemia; MDS, myelodysplastic syndrome; ELN, European LeukemiaNet; CR, complete remission; HCT, hematopoietic-cell transplantation; MRD, minimal/measurable residual disease; qPCR, quantitative polymerase chain reaction.

**Table 1 cancers-15-01658-t001:** Chemotherapeutic agents implicated in t-AML.

Drug Class	Mechanism of Action
Alkylating Agents Cyclophosphamide, * cisplatin, carboplatin, melphalan, busulphan, chlorambucil, lomustin, carmustine, dacarbazine, procarbazine, thiotepa, mitomycin C	Creation of bonds in one or both DNA strands, through alkylation
Topoisomerase II Inhibitors Etoposide, teniposide, doxorubicin, idarubicin, daunorubicin, mitoxantrone, * actinomycin D, amsacrine	“Topoisomerase II poisons” convert topoisomerase II into a DNA-damaging enzyme
Antimetabolites Fludarabine, cladribine, * pentostatin, thiopurines (6-mercaptopurine, * 6-thioguanine, azathioprine *), mycophenolate mofetil *	They act as mimics of other molecules, and in this way, they interfere with DNA and RNA synthesis
Antitubulin Agents Vinblastine, vindesine, vincristine, docetaxel, paclitaxel	Antimitotic agents that bind tubulin dimers and disrupt the formation of mitotic spindle
Poly (ADP-Ribose) Polymerase (PARP) Inhibitors Olaparib, talazoparib, niraparib, rucaparib	Inhibitors of the PARP family of enzymes inhibit homologous recombination repair (PARP enzymes, activated by DNA damage, repair the single-helix DNA breaks by forming branched PAR chains that serve as a docking platform for DNA repair enzymes)

Abbreviations: t-AML, therapy-related acute myeloid leukemia; PARP, poly (ADP-ribose) polymerase. * Widely used as immunosuppressive agents in the treatment of autoimmune and immune-mediated inflammatory disorders.

**Table 2 cancers-15-01658-t002:** Current stratification of de novo and therapy-related acute myeloid leukemia (2022 European LeukemiaNet [ELN] genetic risk classification). For patients with an estimated risk of relapse >35–40% (i.e., patients with adverse and the majority of patents with intermediate-risk profile), allogeneic hematopoietic stem-cell transplantation is indicated.

Favorable
t(8;21) (q22;q22.1); *RUNX1*–*RUNX1T1* inv(16) (p13.1q22) or t(16;16) (p13.1;q22); *CBFB*–*MYH11* Mutated *NPM1* without *FLT3*-ITD bZIP in-frame mutated *CEBPA*
**Intermediate**
Mutated *NPM1* with *FLT3*-ITD Wild-type *NPM1* with *FLT3*-ITD (without adverse-risk genetic lesions) t(9;11) (p21.3;q23.3); *MLLT3*–*KMT2A* Cytogenetic and/or molecular abnormalities not classified as favorable or adverse
**Adverse**
t(6;9) (p23.3;q34.1); *DEK*–*NUP214* t(v;11q23.3)/*KMT2A* rearranged t(9;22) (q34.1;q11.2); *BCR*–*ABL1* t(8;16) (p11;p13); *KAT6A*–*CREBBP* inv(3) (q21.3;q26.2) or t(3;3) (q21.3;q26.2); *GATA2*–*MECOM (EVI1)* t(3q26.2;v)/*MECOM (EVI1)*-rearranged −5 or del(5q); −7; −17 or abnl(17p) Complex karyotype, monosomal karyotype Mutated *ASXL1*, *BCOR*, *EZH2*, *RUNX1*, *SF3B1*, *SRSF2*, *STAG2*, *U2AF1*, or *ZRSR2* Mutated *TP53*

**Table 3 cancers-15-01658-t003:** Biomarker-driven treatment approaches for patients with t-AML.

Biomarker	Treatment Approach
CBF t-AML	“3 + 7” + GO *
t(15;17) t-APL	ATRA + ATO for low-risk; standard treatment or ATRA + ATO + GO for high-risk
*FLT3*-ITD/TKD t-AML	“3 + 7” + midostaurin * Gilteritinib monotherapy Gilteritinib/Ven ^†^ Gilteritinib/Aza ^†^ Aza/Sorafenib ^†^
*TP53*-mutated t-AML	CPX-351 * HMA/Ven Dec × 10 days ^†^
t-AML with MR gene mutations (*SRSF2*, *SF3B1*, *U2AF1*, *ZRSR2*, *ASXL1*, *EZH2*, *BCOR*, *STAG2*)	CPX-351 * HMA/Ven (Dec/Ven for *SF3B1* mutations)
*NPM1*^mut^ t-AML	“3 + 7” ± GO * Aza/Ven ^‡^
Complex-karyotype t-AML	CPX-351 * HMA/Ven
*IDH1*-mutated t-AML	Aza/Ven ^‡^ Ivosidenib Aza/Ivosidenib ^†^
*IDH2*-mutated t-AML	Aza/Ven ^‡^ Enasidenib Aza/Enasidenib ^†^

Abbreviations: t-AML, therapy-related acute myeloid leukemia; t-APL, therapy-related acute promyelocytic leukemia; CBF, core-binding factor; HMA, hypomethylating agent; Aza, azacitidine; Dec, decitabine; Ven, venetoclax; GO, gemtuzumab ozogamicin; *FLT3*-ITD/TKD, *FLT3* with internal tandem duplications/tyrosine kinase domain mutations; MR,-related. * Patients fit for intensive chemotherapy; ^†^ Not approved by regulatory authorities; ^‡^ This mutation predicts a high response rate to Aza/Ven.

**Table 4 cancers-15-01658-t004:** Selected newer agents in clinical development for the treatment of AML including patients with therapy-related AML.

Agent	Mechanism of Action	Indication	Study Design	Study Phase	Trial-Registration Number
Entospletinib	Syk inhibitor	*NPM1*^mut^ AML	“3 + 7” vs. “3 + 7” + entospletinib	Phase 3	NCT05020665
Magrolimab	Blocks CD47 interaction with its ligand SIRPα on phagocytic cells (macrophages), leading to phagocytic elimination of cancer cells	*TP53*^mut^ AML	Aza/Ven vs. Magrolimab + Aza/Ven	Phase 3	NCT05079230
Eprenetapopt (APR246)	p53 protein reconformation/reactivation to restore its proapoptotic and cell-cycle arrest functions	*TP53*^mut^ MDS/AML	Aza vs. APR246 + Aza	Phase 3	NCT03745716
ASTX727	Inhibitor of cytidine deaminase (CDA)	AML in older patents	Dec vs. cedazuridine + Dec	Phase 3	NCT03306264
Galinpepimut-S	WT1 inhibitor	Maintenance AML in CR2	Best available treatment (BAT) vs. Galinpepimut-S + BAT	Phase 3	NCT04229979
Sabatolimab	Anti-TIM-3 antibody	High-risk MDS and AML	Aza/Ven vs. Sabatolimab + Aza/Ven	Phase 2	NCT04150029
Cusatuzumab	Anti-CD70 antibody	AML unfit for intensive chemotherapy	Aza vs. Cusatuzumab + Aza	Phase 2	NCT04023526
Flotetuzumab	Bispecific dual affinity retargeting (DART) antibody-based molecule to CD3ε and CD123	Relapsed/Refractory AML		Phase 1	NCT02152956
Ziftomenib (KO-539)	Menin inhibitor—disrupts the interactions between menin and MLL1 or MLL1-fusion protein; inhibits leukemogenic homeobox A9 (HOXA9) and its cofactor MEIS1 in myeloid stem progenitor cells	*KMT2A* (*MLL*) -rearranged AML and *NPM1*-mutated AML		Phase 1	NCT04067336
Uproleselan	E-selectin inhibitor (targeting bone marrow niche)	Relapsed/Refractory AML		Phase 1	NCT02306291
GTB-3550/GTB-3650	CD33/CD16 bispecific antibody	Relapsed/Refractory AML and high-risk MDS		Phase 1	NCT03214666
AB8939	Tubulin polymerization inhibitor	Relapsed/Refractory AML		Phase 1	NCT05211570
BP1002	Liposomal Bcl-2 antisense oligodeoxynucleotide	Relapsed/Refractory AML		Phase 1	MCT05190471

Abbreviations: AML, acute myeloid leukemia; MDS, myelodysplastic syndrome; Aza, azacitidine; Dec, decitabine; Ven, venetoclax.

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
