# Peer review of "Progress toward Better Treatment of Therapy-Related AML"

_cancers, 2023, doi:10.3390/cancers15061658_

Round 1
Reviewer 1 Report
I find this manuscript to be an excellent review of a significant clinical issue. It is well written and informative for both close participants of the field and for a wider, more general scientific readership. It describes both the etiology of the disease and the (known to date) underlying molecular basis for it. I have learned a good deal from studying it.
I enthusiastically endorse its publication without modification.
Author Response
We would like to express our gratitude for your positive comments.
Reviewer 2 Report
Cancer chemotherapy/radiotherapy-derived AML comprised a good portion of AML cases. The current manuscript by Kotsiafti et.al. systematically reviewed the etiology, pathophysiology and emerging treatment of tAML. Overall, the article is well-comprised and very comprehensive. It is easy to read and will be of interest to many people in the field. I highly recommend the publication of this manuscript.
Author Response

(The authors gave the same response as above.)

Reviewer 3 Report
The development of acute myeloid leukemia following the treatment with chemotherapy or ionizing radiotherapy is a complicated and important problem. The manuscript "Progress toward better treatment of therapy-related AML" describes the causes and prognosis of therapy-related AML and the choice of different treatments considering the variable genetic backgrounds of patients. The manuscript is completely within the scope of Cancers. The manuscript is well written and well illustrated. All major issues related to the mechanisms and treatment strategies of AML are comprehensively described, the cited literature is relevant and modern. In my opinion, this is a high-quality review paper ready for publication.
Author Response

(The authors gave the same response as above.)
